

# RAPiD: a rapid and accurate plant pathogen identification pipeline for on-site nanopore sequencing

Stephen Knobloch[1,2], Fatemeh Salimi[1,3], Anthony Buaya[1], Sebastian Ploch[1] and Marco Thines[1,3,4]

[1] Senckenberg Biodiversity and Climate Research Centre, Senckenberg Society for Nature Research, Frankfurt, Germany
[2] Department of Food Technology, Fulda University of Applied Sciences, Fulda, Germany
[3] LOEWE Centre for Translational Biodiversity Genomics, Frankfurt, Germany
[4] Department of Biological Sciences, Institute of Ecology, Evolution and Diversity, Goethe University Frankfurt, Frankfurt, Germany

Corresponding authors
Stephen Knobloch,
stephen.knobloch@lt.hs-fulda.de
Marco Thines,
marco.thines@senckenberg.de

## ABSTRACT

Nanopore sequencing technology has enabled the rapid, on-site taxonomic identification of samples from anything and anywhere. However, sequencing errors, inadequate databases, as well as the need for bioinformatic expertise and powerful computing resources, have hampered the widespread use of the technology for pathogen identification in the agricultural sector. Here we present RAPiD, a lightweight and accurate real-time taxonomic profiling pipeline. Compared to other metagenomic profilers, RAPiD had a higher classification precision achieved through the use of a curated, non-redundant database of common agricultural pathogens and extensive quality filtering of alignments. On a fungal, bacterial and mixed mock community RAPiD was the only pipeline to detect all members of the communities. We also present a protocol for in-field sample processing enabling pathogen identification from plant sample to sequence within 3 h using low-cost equipment. With sequencing costs continuing to decrease and more high-quality reference genomes becoming available, nanopore sequencing provides a viable method for rapid and accurate pathogen identification in the field. A web implementation of the RAPiD pipeline for real-time analysis is available at https://agrifuture.senckenberg.de.

# INTRODUCTION

Pathogens cause substantial losses in agriculture, causing economic damage and threatening global food security (*Savary et al., 2019*). The transmission of pathogens is accelerated by global trade in agricultural products (*McDonald & Stukenbrock, 2016*) and changing climate patterns which enable the spread of pathogens into regions previously unfavourable to their propagation (*Shaw & Osborne, 2011*). Detecting, identifying and tracking pathogens is therefore critical for preventing or treating disease outbreaks. Identification of agricultural pathogens is traditionally performed by symptom observation and laboratory cultivation of putative pathogens (*Ray et al., 2017*). However, the accuracy

of such methods depends on the skill and experience of the person making the diagnosis (*McCartney et al., 2003*). Other techniques include immunological assays such as enzyme-linked immunosorbent assay (ELISA) and PCR amplification of nucleic acid sequences (*Brunner, Farnleitner & Mach, 2012*), or newer methods such as digital droplet PCR (ddPCR), Loop-Mediated Isothermal Amplification (LAMP) and the use of microfluidic devices (*Patel et al., 2022*). Whereas these can be fast and simple methods for plant pathogen identification, they need to be optimised for each pathogen in question and require specially-trained personnel and equipment. Whole shotgun-metagenomic sequencing on the other hand, is agnostic towards the pathogen and can detect virtually any organism in a sample if sequencing depth is sufficient. The advent of high-throughput metagenomic sequencing has added the benefit of collecting genetic information of pathogens, enabling the tracking of genetic variation and pathogenicity as pathogens spread (*Aragona et al., 2022*). However, this approach has been time-consuming and has required access to specialised facilities and high capital cost equipment, preventing their widespread adoption in pathogen detection and tracking. This limitation has been challenged by the third generation of sequencing devices, most notably Oxford Nanopore Sequencing (ONT) devices, which enable the sequencing of complex microbial communities *in situ* and in less than a day (*Kerkhof, 2021*). Nanopore sequencing has been applied in the medical field, as exemplified by its use in the detection and tracking of SARS-CoV-2 (*Bull et al., 2020*) and the Ebola virus (*Quick et al., 2016*), or as pathogen surveillance tool in clinical settings (*Sanderson et al., 2018*; *Sheahan et al., 2019*). First trials have been conducted for the sequencing of animal viruses (*Jia et al., 2019*) and various plant pathogens (*Bronzato Badial et al., 2018*; *Chalupowicz et al., 2019*; *Boykin et al., 2019*; *Faino et al., 2021*). Yet its widespread adoption in agriculture is still in its infancy. To date, the limitations of this plant pathogen detection technique include the lack of suitable analysis pipelines that can be operated by non-professionals using standard laptop computers, a low specificity of taxonomic assignment and the lack of curated databases of reference pathogen genomes (*Hu et al., 2019*; *Phannareth et al., 2020*; *Ciuffreda, Rodríguez-Pérez & Flores, 2021*). Rapid and *in situ* pathogen detection would not only be limited to the detection of pathogens on infected plant samples in agricultural settings, but could also provide critical information about the presence of pathogens in soils, seeds and the air, thereby becoming a valuable tool for plant protection officers, plant breeders and custom officials.

The aim of this articles is to present the RAPiD pipeline, which combines a lightweight metagenomic classification pipeline with a curated genomic pathogen database that runs locally on a standard laptop computer and enables rapid pathogen detection under field conditions without prior bioinformatic training. To evaluate its performance against other tools, three mock communities were investigated and classification results analysed. In addition, a fast sample-to-sequence workflow for processing plant samples in the field is presented.

## MATERIALS AND METHODS

### Pipeline description

The RAPiD pipeline (Fig. 1) (https://github.com/SteveKnobloch/RAPiD_pipeline) was developed in Nextflow (*Di Tommaso et al., 2017*) and containerised with Docker (*Merkel, 2014*). In the first step, basecalled reads are quality filtered using Porechop (*Wick, 2017*) to remove adapters, followed by processing in NanoFilt and NanoLyse (*De Coster et al., 2018*) to remove low quality (q < 8) and small (<1,000 bp) reads, as well as reads matching the Lambda phage control genome (GenBank accession J02459.1). Quality filtered reads are then aligned to an indexed reference database (see section on reference database construction) using minimap2 version 2.21 (*Li, 2018*) with the *-ax map-ont* flag and without secondary alignments. Alignments are then filtered using SAMtools (*Li et al., 2009*) to remove supplemental alignments and alignments with a DP alignment score (*AS*) below 2,000, less than 200 minimizers on the chain (*cm*) or a gap-compressed per-base sequence divergence (*de*) above 0.075. These values were determined suitable to filter the majority of non-target sequences at the species level (see results section on cut-off value estimation). Filtered alignments are then summarised by species or subspecies rank and a mean normalised alignment score $\left( \dfrac{AS}{sequence\ length} \times 50 \right)$ and mean per base identity score $((1 - de) \times 100)$ calculated for each taxonomic group. The results are output into an alignment and report summary. For simplifying interpretation of the report, a confidence value is provided for each taxonomic group based on the number of hits and the mean normalised alignment score, with at least two hits and a score above 75 considered "high", two hits and a score between 70 and 75 considered "medium" and one hit or a score below or equal 70 considered "low". Additional features of the web-based application are live base calling using guppy (available from https://community.nanoporetech.com) and automatic expansion of the reference database with user-supplied reference genomes (Fig. 1).

### Cut-off value estimation

A reduced reference index size is necessary to improve computational speed and memory requirements of the pipeline, but enables false positive matches due to a lack of off-target reference genomes and presence of highly conserved genes across taxa. To reduce the rate of false positive matches, suitable alignment score cut-off values were determined based on the alignment of publicly available whole genome nanopore sequences, consisting of 14 pairs of closely related species, against a reference index containing only the genome of one species of each pair (see Table S1). Due to differences in sequence quality between datasets, a wide range of species were chosen for which publicly available Nanopore data was available. These included species of the genera *Pseudomonas*, *Burkholderia*, *Spiroplasma*, *Lactobacillus*, *Xanthomonas*, *Clostridium*, *Vibrio*, *Staphylococcus*, *Puccinia*, *Fusarium*, *Phytophthora*, *Bipolaris*, *Colletotrichum* and *Penicillium*. Raw genomic reads for each BioSample were downloaded from the NCBI SRA using the sratoolkit (*Leinonen, Sugawara & Shumway, 2011*). Reads smaller than 1,000 bp were removed with NanoFilt

 

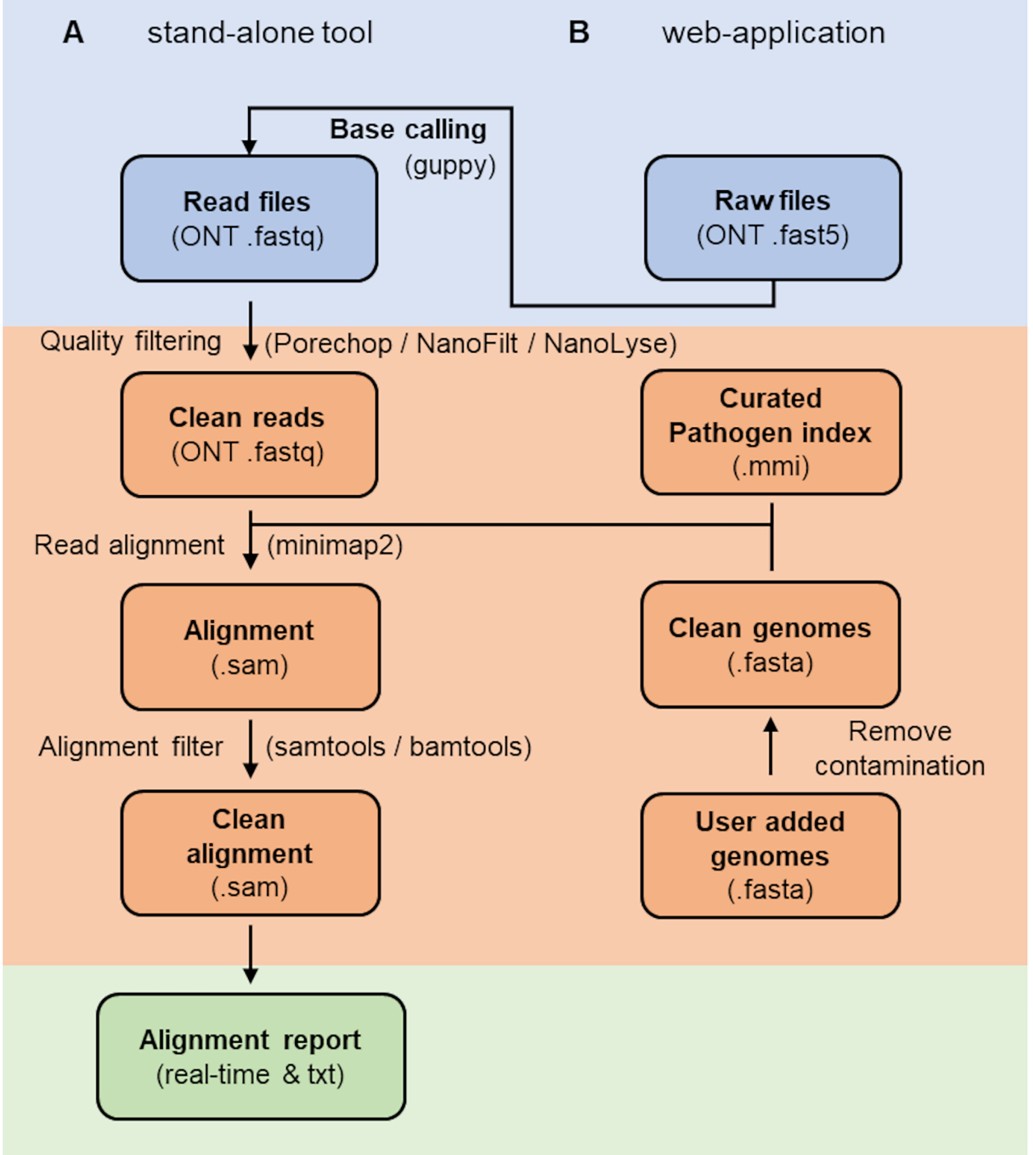

**Figure 1 Workflow of the RAPiD pipeline.** Workfow of the RAPiD pipeline, divided into data input (blue), read processing, index creation and read alignment (orange), and report generation (green). The web-application enables additional basecalling of raw fast5 files and extension of the reference database (pathogen index) through submission of new genome assemblies.

and each resulting dataset normalised to 100,000 reads using the reformat.sh script of the BBMap tool (*Bushnell, 2014*). A reference index was built in minimap2 using the corresponding representative genomes on NCBI GenBank for one species of each pair. All raw reads were then mapped against the index as detailed above.

## Reference database construction and contaminant removal

To construct a non-redundant database for minimap2 alignment within the RAPiD pipeline, representative genomes of 190 plant pathogenic bacteria (13 genera), fungi (76

genera) and oomycetes (eight genera) (Table S2) were downloaded with genome_updater version 0.2.5 (available at https://github.com/pirovc/genome_updater) from NCBI GenBank (accessed on 05.05.2022). Sequences smaller than 2,000 bp were removed. Mitochondrial genomes were removed by performing a BLASTN (Altschul et al., 1990; Camacho et al., 2009) query against the NCBI mitochondrion dataset (https://ftp.ncbi.nlm.nih.gov/refseq/release/mitochondrion/, accessed 15.05.2022). In total, 661 contigs with a percent identity above 80% to a mitochondrial genome, a coverage of at least 500 bp and a subject size below 200,000 bp were removed. Cross-kingdom contamination was checked using Conterminator (Steinegger & Salzberg, 2020). In brief, all-against-all alignments in Conterminator were performed with all representative plant genomes and the representative human genome from NCBI RefSeq (accessed on 01.06.2022). In total, 2,476 sequences that matched a different kingdom than the original accession were removed. Additional, manual inspection of long sequences, that did not get flagged in Conterminator due to their size, was performed using BLASTN against the NCBI nt database. Extensive removal of contaminants was important as representative genomes such as *Pseudoperonospora cubensis* (GenBank accession AHJF00000000.1) contained sequences larger than 49,000 bp of *Cucumis sativus* and *Acinetobacter calcoaceticus* which would otherwise flag as positives in an uninfected sample. A list of all removed sequences is compiled in Table S3. The filtered database was then indexed using minimap2 with the parameter −I 2G. For sub-species identification, the same procedure for index creation was performed, apart from that all genomes corresponding to the 190 plant pathogenic bacteria, fungi and oomycetes were downloaded from NCBI GenBank.

## Mock community construction

Two *in silico* mock communities and one biological mock community were constructed to compare the classification results of the RAPiD pipeline against other available tools. The first *in silico* community consisted of publicly available nanopore sequences from 15 individual bacteria, fungi and oomycetes (see Table S4) downloaded with sratools from the NCBI SRA database. Each dataset was quality filtered before analysis, using Porechop to remove adapters, NanoFilt to remove reads smaller than 1,000 bp and below a quality score of eight, NanoLyse to remove reads aligning to the Lambda reference genome and BBMap to subset each dataset to 5,000 reads. The second *in silico* mock community consisted of nanopore sequences from 11 pathovars or strains in the *Pseudomonas syringae* group (see Table S4) to test for sub-species identification. Quality filtering and subsetting was performed as mentioned above, apart for dataset SRR9943113, which only contained 1,219 reads after quality filtering and dataset SRR10431754, which consisted of four different strains and was thus subset to 20,000 reads. The biological mock community consisted of nine *Fusarium* species (see Table S4). DNA was extracted from each strain using a modified CTAB-extraction, quantified using a Qubit dsDNA HS assay (Thermo-Fisher Scientific, Waltham, MA, USA) and pooled in equal concentrations by adding 200 ng gDNA of each organism to a tube. A sequence library was constructed using the LSK110 kit (Oxford Nanopore Technologies, Oxford, UK) according to the manufacturer's instructions and sequenced on a R9.4.1 Flongle flow cell connected to a MinION Mk1C

device (Oxford Nanopore Technologies, Oxford, UK). The sequencing run generated 131.17 k reads with an N50 of 14.61 kbp. Raw sequences were basecalled with guppy version 6.1.7 and quality filtering was performed as for the other mock communities. No subsetting of read numbers was conducted, leaving a total of 85,532 reads for analysis.

## Comparison of RAPiD to other metagenomic classification tools

The three mock communities were analysed with: (1) the RAPiD pipeline; (2) BugSeq (*Fan, Huang & Chorlton, 2021*), a web-based and commercial tool; (3) WIMP (*Juul et al., 2015*), the classification pipeline of Oxford Nanopore Technologies which uses centrifuge (*Kim et al., 2016*) as classification engine; (4) Kraken2 version 2.1.2 (*Wood, Lu & Langmead, 2019*), a short-read classifier; and (5) an approach using minimap2 alignment against the entire NCBI nt database followed by MEGAN (MEGAN6 Community Edition) lowest common ancestor calculation (*Huson et al., 2016*). For analysis with BugSeq (accessed February 2022), each mock community was uploaded and analysed against the NCBI nt database. Classification with WIMP was performed by uploading each mock community through the EPI2ME agent (Oxford Nanopore Technologies, Oxford, UK) and choosing Fastq WIMP analysis against the database v2021.11.26. Classification with Kraken2 was performed with default parameters against the precompiled database PlusPF and the memory reduced database PlusPF-8 retrieved from https://benlangmead.github.io/aws-indexes/k2 (accessed July 2023). For the minimap2-MEGAN pipeline, first the NCBI nt database was downloaded (http://ftp.ncbi.nlm.nih.gov/blast/db/FASTA/nt.gz, accessed February 2022) and indexed using minimap2 with default parameters. Alignment of each mock community was then performed against the index with minimap2 and the *-ax map-ont* flag. The alignment file was subsequently subject to lowest common ancestor (LCA) calculation using the MEGAN sam2rma script with the *-lg -alg longReads* flags against the MEGAN mapping file megan-nucl-Jan201.db (https://software-ab.cs.uni-tuebingen.de/download/megan6/welcome.html, accessed January 2022).

Read tables were then summarised by counting reads at species or sub-species level. Reads that matched a target species or sub-species present in the mock community were classified as true positives. In contrast, reads that matched a species or sub-species not present in the mock community were classified as false positives. For each mock community, the precision, recall and F1 score was calculated along with the number of true and false taxa detected with each tool. Precision was calculated as the ratio between true positive reads and all reads classified at the species or sub-species level. Recall was calculated as the ratio between true positive reads and all reads analysed. The F1-score was calculated as $2 * (precision * recall) / (precision + recall)$. For the *Pseudomonas syringae* mock community, reads not assigned to the correct pathovar or strain were considered false positives.

The Kraken2 pipeline with the PlusPF database and the minimap2-MEGAN approach were run on one node of an HPC (five nodes, each with 64 cores (AMD EPYC 7301) and 1 TB RAM). The RAPiD pipeline and Kraken2 with the PlusPF-8 database were run on a laptop (Dell Latitude 5420, Intel Core i5-1135G7 and 16 GB RAM). With the exception of

the web-based tools, runtime, CPU-time and maximum memory requirements were recorded for each analysis.

## In-field sample collection, DNA extraction and sequencing

The following protocol was developed to rapidly process samples for sequencing outside of the laboratory without the use of dangerous chemicals in the DNA extraction procedure or specialised equipment, such as centrifuges, thermocyclers and heat blocks. The set-up for in-field sample processing and sequencing is depicted in Fig. 2. Samples of wheat, tomato and apple leaves showing signs of infection were collected and processed individually as follows. Approximately 100 mg of sample was cut into small pieces with a sterile scalpel, transferred to a 1.5 ml Eppendorf tube and immediately macerated in 800 µl of CTAB-Buffer (2% CTAB, 2% PVP, 100 mM Tris, 1.4M NaCl, 20 mM $Na_2EDTA$, pH 8) with 14 µl of 20 g $l^{-1}$ Proteinase K using a hand-held pestle grinder (DWK Life Sciences Kontes). The sample was then kept at approximately 65 °C for 15 min using a thermal flask and glass thermometer. The lysate was then subject to DNA purification with a S-TECH device (DNAiTECH) according to the manufacturer's instructions for soil samples and resuspended in 140 µl of elution buffer (10 mM Tris-HCl pH 8.5). The resuspended DNA was then normalised to approximately 500 ng in 23.5 µl elution buffer with a pre-made, modified concentration of MAG-BIND TotalPure NGS magnetic beads (Omega Bio-tek). In short, 75 µl of MAG-BIND TotalPure NGS magnetic beads was thoroughly resuspended and 3.5 µl transferred to a new tube. The remaining reagent was spun-down to pellet the beads and 66.5 µl of the supernatant was added to the previously aliquoted 3.5 µl of beads. This resulted in 70 ul of binding reagent with a reduced number of beads for a 0.5x clean-up and concentration of the DNA. In previous experiments, this concentration of beads bound and subsequently released approximately 400–600 ng of DNA in 23.5 µl of elution buffer from various plant samples (data not shown). All subsequent clean-up steps were performed according to the manufacturer's instructions. Sequence libraries were constructed using the LSK114 kit according to the manufacturer's instructions with omission of the first magnetic bead clean-up step. The temperature-controlled steps were achieved by adding hot or cold water to a thermal flask with a glass thermometer. Libraries were sequenced on R10.4.1 Flongle flow cells on a battery-operated MinION Mk1C device (Oxford Nanopore Technologies, Oxford, UK). Basecalling was performed in simplex high accuracy (HAC) mode. RAPiD was run on a laptop with 16 Gb RAM. The workflow from sample to DNA library took less than 2 h per sample. The total runtime for each sequencing run was dependent on the battery pack and was approximately 5.5 h. The DNA concentration for each sample after DNA extraction was analysed in the laboratory after the field trial using a Qubit dsDNA HS assay and is listed in Table S5.

## Data availability

Raw sequence data of the *Fusarium* spp. mock community and the three infected plant datasets sequenced in the field are stored in the NCBI SRA under BioProject PRJNA1063731.

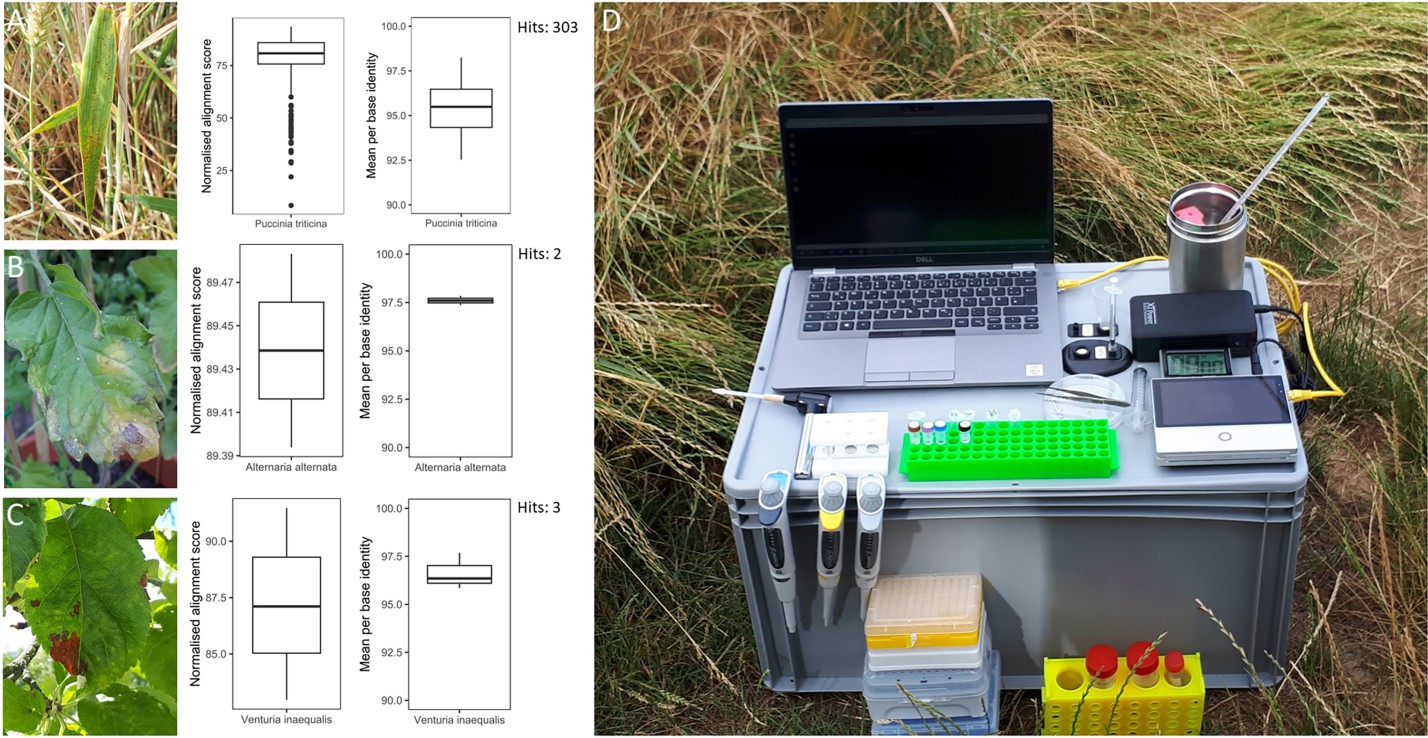

**Figure 2 Overview of in-field plant pathogen detection with necessary equipment.** Pictures of infected plant samples with normalised alignment score and mean per base identity for sequence matches: (A) common wheat plant with wheat leaf rust (*Puccinia triticina*); (B) tomato plant with leaf blight (*Alternaria* spp.); (C) apple plant leaf with apple scab (*Venturia inaequalis*); (D) instrument set-up for DNA extraction and sequencing in the field on an Oxford Nanopore Technology Mk1C device.   

## RESULTS

### Cut-off value estimation

The values for alignment metrics of target and non-target species against the reference database are shown in Fig. 3. The metrics most discriminative of a true positive match were the DP alignment score (*AS*), number of minimizers on the chain (*cm*) and the gap-compressed per-base sequence divergence (*de*). The chaining score (*s1*) largely overlapped with *cm*. Despite not having a reference genome in the database, each non-target species matched a false reference genome. Different cut-off values were compared to exclude most false positives while retaining most true positive reads. Different cut-off values were investigated to find a setting that retained the majority of target reads while excluding over 99% of off-target reads. Choosing values of 2,000, 200 and 0.075 for *AS*, *cm* and *de*, respectively, included 57.98 ± 23.95% of all mapped target reads while excluding 99.18 ± 1.93% of all off-target reads. Upon closer examination, many remaining false positive reads matched conserved genes of the mitochondrial genome, the ribosomal RNA genes or internal transcribed spacer regions. Other alignments metrics such as mapping quality (*MAPQ*) and total number of mismatches and gaps (*NM*) were less suitable for species-level discrimination (Fig. 3).

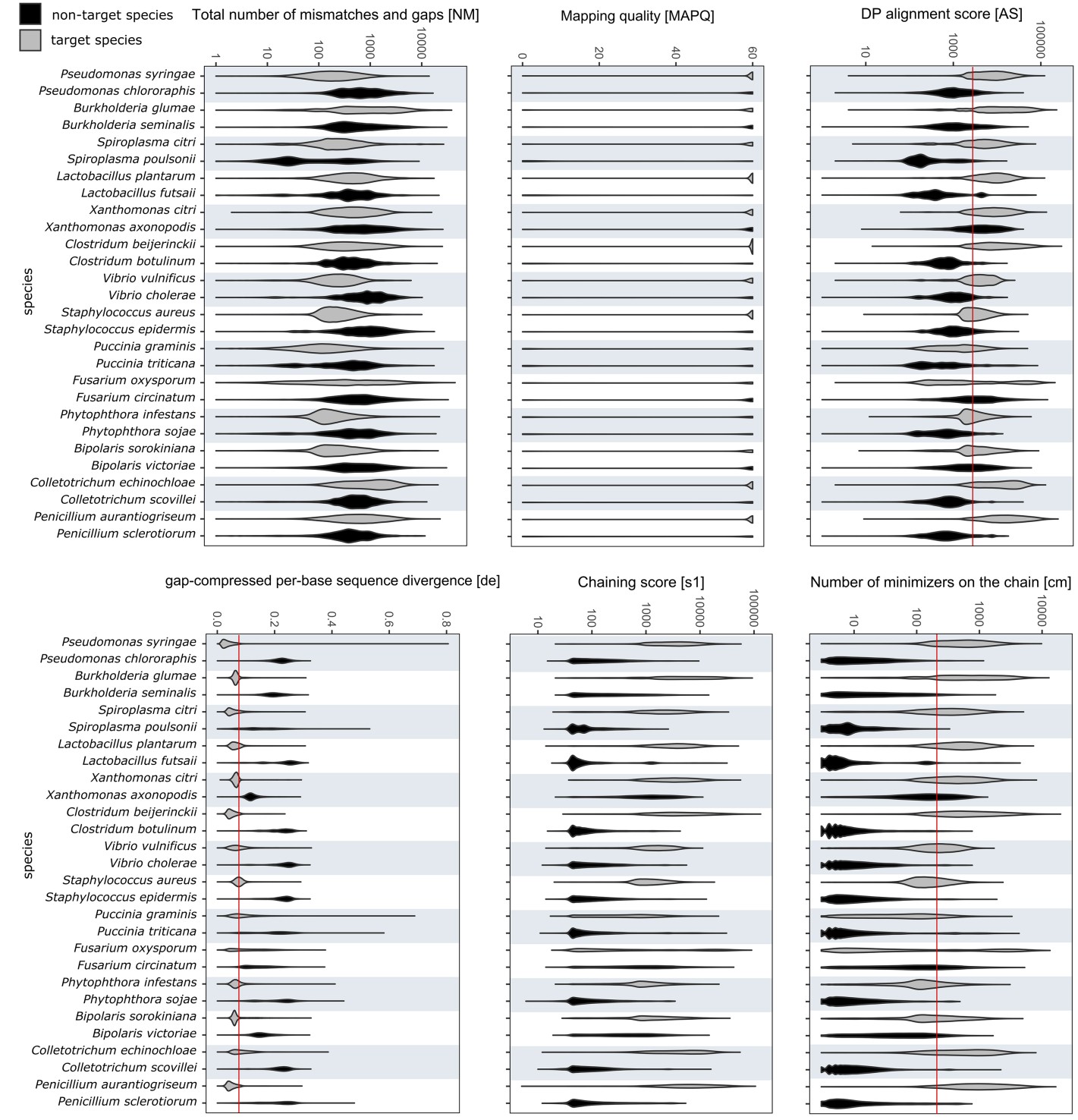

**Figure 3 RAPiD pipeline values for alignment metrics against the reference database.** Values for alignment metrics of target (grey) and non-target species (black) against a reference database containing representative genomes of each target species. Cut-off values excluding over 99% of false positive matches are shown as red lines in selected metrics.

## Mock community analysis

The RAPiD pipeline had the highest precision for each of the mock communities, with 0.99, 0.94 and 0.98 for the mixed mock, *P. syringae* mock and *Fusarium* mock, respectively (Table 1). In addition, it detected all taxa in the mock correctly. High precision, however, came at the cost of lower recall of 0.50, 0.50 and 0.37, respectively. RAPiD had the highest F1 scores of 0.65 for the *P. syringae* mock and 0.53 for the *Fusarium* mock. For the mixed mock, RAPiD had a lower F1 score than all pipelines apart from WIMP. BugSeq correctly assigned all 15 species of the mixed mock with a precision of 0.94 and recall of 0.80. However, it did not detect five pathovars of the *P. syringae* mock and seven species of the *Fusarium* mock, leading to a lower precision of 0.33 and 0.44, respectively. WIMP detected seven of the species in the mixed mock, six pathovars in the *P. syringae* mock and none of the correct species in the *Fusarium* mock. The precision was 0.68, 0.51 and 0 for each of the mock communities, respectively. The Kraken2 pipeline performed similarly between both the precompiled, large and memory-reduced databases, detecting 10 species in the mixed mock and seven pathovars in the *P. syringae* mock, leading to a precision of between 0.26 and 0.88 and recall of 0.22 and 0.61, respectively. Similar to WIMP, none of the *Fusarium* species were detected with Kraken2. The minimap2-MEGAN pipeline performed best on the mixed mock community, detecting all species with a precision of 0.95. On the *P. syringae* and *Fusarium* mocks it detected 6 and 5 taxa with a precision of 0.24 and 0.31, respectively.

The number of taxa wrongly assigned was highest for WIMP and Kraken2, which assigned reads to over 1,000 species not in the mock community. The pipeline with the lowest number of wrongly assigned taxa was the minimap2-MEGAN approach with 7 to 25 taxa depending on the mock community, followed by the RAPiD pipeline with 9 to 82 wrongly assigned taxa. Runtime was shortest for Kraken2 with the 8GB database, followed by Kraken2 with the full reference database and then WIMP, RAPiD and BugSeq. The memory requirements were lowest for Kraken2 with the 8GB database, followed by RAPiD, the minimap2-MEGAN approach and Kraken2 with the full reference database.

## Pathogen detection in the field

Three infected plant samples were processed and their DNA sequenced in the field. These included a leaf of a common wheat plant with wheat leaf rust (*Puccinia triticina*), a leaf of a tomato plant with leaf blight (*Alternaria* spp.) and a leaf of an apple tree with apple scab (*Venturia inaequalis*) (Fig. 2). The instrument set-up for DNA extraction and sequencing is shown in Fig. 2. DNA concentrations of the three plant samples ranged from 4.5 to 12.2 ng $\mu l^{-1}$ (Table S5). The time from sample collection to starting the sequencing run took less than 2 h. Sequencing generated 68.7 to 192.6 Mbp of raw sequences or 37,565 to 117,610 reads per sample (Table S5). After sequencing commenced, the first sequence match for *P. triticina* on the wheat sample was detected within 30 min. In total, 303 sequences were assigned to *P. triticina* out of 16,027 reads that passed quality filtering and the average normalized alignment score for the matches was 77.48 with a mean per base identity of 95.37% (Fig. 2). For the tomato sample, two hits matched *Alternaria alternata* out of 27,853 quality filtered reads. The normalized alignment score for these matches was

**Table 1 Comparison of the RAPiD pipeline against different metagenomic classifiers.**

| Dataset | Method | Run-time | Max. memory | Classified reads ≥ species level | % True positive reads (taxa) | % False positive reads (taxa) | Precision | Recall | F1 |
|---|---|---|---|---|---|---|---|---|---|
| 15 Genomes mixed mock | BugSeq | 02:15 h | N/A | 63,673 | 93.7 (15) | 6.3 (148) | 0.94 | **0.80** | **0.86** |
| 75,000 Reads | WIMP | 00:26 h | N/A | 43,299 | 67.7 (7) | 32.3 (1,303) | 0.68 | 0.39 | 0.50 |
| | Kraken2 full DB | 00:05 h | 74.86 | 58,100 | 79.3 (10) | 20.7 (1,953) | 0.79 | 0.61 | 0.69 |
| | Kraken2 8GB DB | **00:02 h** | **7.95** | 48,745 | 88.3 (10) | 11.7 (1,116) | 0.88 | 0.57 | 0.70 |
| | minimap2 MEGAN nt | 23:20 h | 27.29 | 51,802 | 95.2 (15) | 4.8 (13) | 0.95 | 0.66 | 0.78 |
| | **RAPiD pipeline** | 00:51 h | 14.04 | 38,213 | 99.0 (15) | 1.0 (18) | **0.99** | 0.50 | 0.67 |
| 11 *P. syringae* pathovars | BugSeq | 01:50 h | N/A | 49,747 | 33.1 (6) | 66.9 (191) | 0.33 | 0.32 | 0.33 |
| 51,219 Reads | WIMP | 00:46 h | N/A | 50,739 | 51.3 (6) | 48.7 (397) | 0.51 | **0.51** | 0.51 |
| | Kraken2_full | 00:05 h | 74.80 | 48,780 | 29.8 (7) | 70.2 (454) | 0.30 | 0.28 | 0.29 |
| | Kraken2_8GB | **00:01 h** | **7.82** | 43,802 | 26.0 (7) | 74.0 (382) | 0.26 | 0.22 | 0.24 |
| | minimap2_MEGAN_nt | 18:12 h | 25.75 | 35,743 | 24.2 (6) | 75.8 (25) | 0.24 | 0.17 | 0.20 |
| | **RAPiD pipeline*** | 06:51 h | 13.78 | 27,012 | 93.9 (11) | 6.1 (83) | **0.94** | 0.50 | **0.65** |
| 9 *Fusarium* spp. | BugSeq | 02:00 h | N/A | 70,553 | 53.3 (2) | 46.7 (51) | 0.44 | **0.44** | 0.44 |
| 85,532 Reads | WIMP | 00:56 h | N/A | 22,019 | 0 (0) | 100 (1,477) | 0.00 | 0.00 | NA |
| | Kraken2_full | 00:05 h | 74.83 | 58,480 | 0 (0) | 100 (546) | 0.00 | 0.00 | NA |
| | Kraken2_8GB | **00:01 h** | **7.92** | 50,831 | 0 (0) | 100 (642) | 0.00 | 0.00 | NA |
| | minimap2_MEGAN_nt | 07:18 h | 27.22 | 50,846 | 31.0 (5) | 69.0 (7) | 0.31 | 0.18 | 0.23 |
| | **RAPiD pipeline** | 01:10 h | 14.01 | 31,730 | 98.4 (9) | 1.6 (9) | **0.98** | 0.37 | **0.53** |

**Note:**
Statistics for mock community analysis with different classification tools. Asterisk indicates where extended RAPiD database, including all reference genomes from GenBank, was used. Scores in bold indicate the best results for each metric and mock community.

89.44 with a mean per base identity of 97.60%. The first hit was detected within 3 h of sequencing. For the apple sample, three sequences were assigned to *V. inaequalis* out of 46,039 reads after quality filtering. The average normalized alignment score was 87.19 with a mean per base identity of 96.63%. The first hit was detected within 1.5 h after sequencing commenced. No false positive hits were recorded for any of the three samples. A BLASTN search of all matching sequences against the nt database (accessed 01.07.24) confirmed the presence of *Puccinia triticina* and *Alternaria alternata*. However, the apple sample sequences showed closer sequence similarity to *Venturia effusa*, albeit with low percent sequence identity, suggesting a different pathogen currently not present in the RAPiD database. All sequence hits are listed in Table S6.

## DISCUSSION

The RAPiD pipeline was developed to provide fast and accurate plant pathogen detection in the field without the need for large computing resources. Using a limited genomic reference database and selecting specific cut-off values for alignment metrics enabled the exclusion of most false positive matches while retaining over half of the true positive hits on a mixed species dataset of nanopore sequences. Other strategies to reduce false positive classification include combining the output of several classification tools (*McIntyre et al., 2017*) or lowest

common ancestor inference (*Nasko et al., 2018*). However, these depend on large reference databases that would be prohibitive for use in real-time analysis in the field. Therefore, the strategy presented here is suitable as a light-weight offline metagenomic classifier of common plant pathogens species. Using metagenomic sequencing, as opposed to amplicon sequencing, further eliminates the need for time-intensive PCR steps and potentially enables more accurate read classifications (*Usyk et al., 2023*). The presence of highly conserved genes and contaminated sequences in publicly available genome databases have both been identified as limitations for correct metagenomic classification (*Marcelino, Holmes & Sorrell, 2020*; *Steinegger & Salzberg, 2020*). The addition of contaminant removal and removal of mitochondrial reads from the RAPiD reference database further reduced the rate of false positive matches, enabling the accurate detection of plant pathogens. Whereas RAPiD only enables species or sub-species matches to taxa that are currently present in the reference database, the reduction of costs for sequencing whole genomes will lead to more reference genomes of plant pathogens becoming available in the future (*Aragona et al., 2022*) and thus extend the scope of the detection pipeline.

Apart from the cloud-based classification tools, only RAPiD and Kraken2 with the memory-reduced reference database, would be able to be run on a normal laptop environment and provide results in real-time while sequencing is ongoing. This would not be possible with the other tools, due to the high memory requirements or run-time. In terms of precision, RAPiD was the only tool to identify all taxa in the mock communities analysed and with a high precision. One of the critical factors in metagenomic classification is the type and scope of the reference database used (*Sczyrba et al., 2017*; *Anyansi et al., 2020*). This study only compared tools with their own native reference databases and as such cannot make a direct comparison between the performances of the tools or algorithms themselves. Rather, it shows that general purpose databases are not always suitable for specialised tasks such as pathogen detection. This is exemplified by an absence of any of the correct *Fusarium* species in the WIMP and Kraken2 databases and of some *P. syringae* pathovars in the other databases. Although precision of the RAPiD pipeline was high, this came at a cost of a lower recall introduced by extensive removal of low-quality alignments. This is, however, less important if only pathogen detection, rather than entire metagenomic classification, is required. While this study shows that sub-species or pathovar detection is possible with the RAPiD pipeline, it remains a difficult task for metagenomic classifiers in general (*Breitwieser, Lu & Salzberg, 2019*; *Anyansi et al., 2020*; *Frioux et al., 2020*) and likely depends on the genetic divergence between strains. This is also highlighted by a higher number of wrongly classified taxa in the *P. syringae* mock community analysed at sub-species level compared to the two mock communities analysed at species-level (Table 1). Being able to identify a specific strain with high confidence will require further research into refining the set of cut-off values to prevent false positive hits. Extending the database to contain more reference genomes to account for sub-species identification (as was the case for RAPiD analysis of the *P. syringae* mock community) also increased the run time (Table 1), which must be taken into account when sequencing in the field.

The presented protocol for on-site plant DNA extraction and sequencing was successful for all three tested crop species. This adds to the list of *in situ* sample preparation and sequencing protocols for other sample types such as animal tissue (*Menegon et al., 2017*; *Pomerantz et al., 2018*), microbial mats (*Johnson et al., 2017*), geothermal hot springs (*Gowers et al., 2019*) or corals (*Carradec et al., 2020*) and provides a basis for standardising the work flow and necessary equipment for a wider variety of plant species in the future. The difference in the number of reads assigned to a pathogen from the three infected plant samples, ranging from two to 303 reads, shows that the ratio of host-to-pathogen reads could be a limitation if sequencing depth is shallow. However, despite an overwhelming proportion of host sequences, the pathogen in question was detectable in each of the three cases. To further enable detection of low-abundant pathogens, future protocols could focus on excluding the majority of host DNA. This could be conducted either during the sample processing steps (*Bag et al., 2016*; *Marotz et al., 2018*) or during sequencing with the adaptive sampling feature on Oxford Nanopore Technologies devices (*Martin et al., 2022*). Sequencing deeper or for longer periods of time would also provide a higher chance of detecting sequences from low abundant pathogens in sufficiently high numbers to make a confident identification possible. The detection of pathogens using reduced, curated reference databases is also limited only to the pathogens present within it. Extending the database with additional pathogens could have an impact on run time and necessitate more powerful computing resources.

### Funding

This work was funded by the Federal Ministry of Food and Agriculture (BMEL) project AGRIFUTURE, grant number 28A8702X19, and REACT-EU project Pandprep, grant number FPG968. The funders had no role in study design, data collection and analysis, decision to publish, or preparation of the manuscript.

### Grant Disclosures

The following grant information was disclosed by the authors:
Federal Ministry of Food and Agriculture (BMEL): 28A8702X19.
REACT-EU project Pandprep: FPG968.

### Competing Interests

The authors declare that they have no competing interests.

### Author Contributions

- Stephen Knobloch conceived and designed the experiments, performed the experiments, analyzed the data, prepared figures and/or tables, authored or reviewed drafts of the article, and approved the final draft.
- Fatemeh Salimi performed the experiments, authored or reviewed drafts of the article, and approved the final draft.

 

- Anthony Buaya performed the experiments, authored or reviewed drafts of the article, and approved the final draft.
- Sebastian Ploch performed the experiments, authored or reviewed drafts of the article, and approved the final draft.
- Marco Thines conceived and designed the experiments, authored or reviewed drafts of the article, and approved the final draft.

### DNA Deposition

The following information was supplied regarding the deposition of DNA sequences:

Raw sequence data of the Fusarium spp. mock community and the three infected plant datasets sequenced in the field are available at the NCBI SRA: PRJNA1063731.

### Data Availability

All data is available in the NCBI SRA: PRJNA1063731.

Code for the RAPiD pipeline is available at Zenodo: Knobloch, S. (2024). RAPiD pipeline (version 1). Zenodo. https://doi.org/10.5281/zenodo.11404180.

### Supplemental Information

Supplemental information for this article can be found online at http://dx.doi.org/10.7717/peerj.17893#supplemental-information.

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
