# Peer review of "RAPiD: a rapid and accurate plant pathogen identification pipeline for on-site nanopore sequencing"

_PeerJ, doi:10.7717/peerj.17893_

## Round 0.1 · original submission · Major Revisions

Dear Dr. Knobloch and colleagues:

Thanks for submitting your manuscript to PeerJ. I have now received two independent reviews of your work, and as you will see, the reviewers raised some concerns about the research (mostly missing information and methodological repeatability). Despite this, these reviewers are optimistic about your work and the potential impact it will have on research studying sequence-based plant pathogen identification in the field. Thus, I encourage you to revise your manuscript, accordingly, considering all of the concerns raised by both reviewers.

Please revise your manuscript for clarity and limit jargon/reduce verbiage but be clear about the focus of the study and the target audience. Focus on especially on clarity, and address the sections considered incomplete and/or unclear by the reviewers. It appears that certain key references are missing. The Methods should be clear, concise and repeatable. Please ensure this. Also, elaborate on the discussion of your findings, placing them within a broad and inclusive body of work by the field. Please supply any code or scripts in the supplemental material.

I look forward to seeing your revision, and thanks again for submitting your work to PeerJ.

Good luck with your revision,

-joe

·

Basic reporting

The manuscript describes the development of a novel pipeline for rapid and accurate plant pathogen identification. Overall it is well written, the introduction is clear, concise, provides sufficient background, and describes the most important knowledge gaps. Additional information about strains and assemblies used is well documented in the supplementary tables. However, regarding “basic reporting”, I have the following comments/suggestions:
- The authors provide some info about alternative plant detection techniques, but refer to a rather old review. There are very recent reviews that also discuss newer techniques than ELISA or PCR. I strongly recommend to use a recent reference also pointing to newer techniques, e.g. ddPCR, LFA, RPA, amplicon sequencing.
- The figure legends should be expanded to allow the reader to fully understand the figure without going to the text. For instance, what is the meaning of the color in figure 1,

Additional minor comments:
- L40: we cannot access the website because it is deemed unsafe by chrome
- The reference list could use some cleaning up, e.g. consistency in adding doi nrs in a correct way (L400, L402, L443,…) or in adding volume/issue nrs (e.g. L395), checking doi urls (e.g. L391)
- Genus and species names should be in Italic (e.g. legend fig 3)
- Define abbreviations at first use (e.g. L196: LCA)
- Perhaps the difference between true and false-positives could be described a bit more clearly.
- Typo’s:
o L208: kraken
o L265: species-level?

Experimental design

The research question is clearly defined and relevant for users of nanopore sequencing technology to detect target microorganisms. The used methodology is generally clearly described. The results are presented in a clear way. I like the way the authors defined cut-off values for accurate identification at species level. I also appreciate their creativity to design the in-field analysis set-up in order to provide proof-of-concept of the use of nanopore sequencing at point of care. Nevertheless, in some cases more details could be provided on the methodology:
- L209: one core seems low for this application, could you mean one computing node instead? Also, although RAM is important, it is quite a meaningless metric by itself. Try and give more detailed specifications of the computing resources used in this paper, as this helps fellow researches to make more educated decisions if they wish to apply this pipeline.
- It is important to provide version numbers of the programs used in this study (e.g. minimap2 has a better memory management system in version 2.21 compared to 2.19)
- The authors clearly describe which alignment parameters were investigated, and which were found suitable for assessing accuracy of identification. However, it is not exactly clear how the authors came up with these exact cutoffs for AS, de and cm.
- The authors make use of a reduced reference index, containing (only) around 190 pathogens. It would be good that the authors also indicate how many genera are represented for bacteria, fungi, and oomycetes. This would give the reader a better idea of how representative this reference database is.
- For the Fusarium mock community: was each species barcoded, or was there some way to discern between each strain after the sequencing run employed? How sure are the authors that a positive hit to a certain strain was indeed due to DNA of that strain and it was not a false-positive hit. It would also be good to present more detailed data on the true positive hits obtained for each Fusarium spp. This should also be equal since equal amounts of DNA were pooled. More information of the sequencing run should also be provided. What was the distribution of lengths? Is there a bias in sequencing of shorter fragments?
- For the field analysis, the nanopore-based identification of the pathogen should be confirmed with a complementary test, e.g. specific qPCRs, or Sanger sequencing of genetic markers? In other words, how can the authors be absolutely sure that the identified pathogen is indeed the cause of the disease symptoms?
- In table 1, the number of true positives and false positives could be presented in % of classified reads to allow more easy comparison between the tested pipelines.
- Rapid uses a curated database (limited number of sequences in the database and also non-informative (mitochondrial) or contaminated sequences are removed. Considering that most other tools do not make use of a simplified reference database, there may be a bias in the results regarding run time as well as in detection of false-positives. Use of similar databases for all pipelines would give a much better comparison between the pipelines.

Additional minor comments:
- L171: pooled in equal concentrations  specify
- Maybe the authors could make a clear distinction between “in silico” and “biological” mock communities. Now using “the first in silico mock community”, “the second in silico mock community”, “the mixed mock community”, and “the third mock community” may be a bit confusing
- L203: ration instead of difference?
- Figure2: the x-axis of the alignment score figure should be more clearly labeled (10, 1000, 100000)
- L292: minimap2-megan is spelled wrong
- L240 and onwards: how were the nanopore reads base-called (FAST,HAC, SUP, duplex?). This can have an impact on accuracy of base-calling and required analysis time

Validity of the findings

The manuscript provides proof-of-concept of the identification pipeline with in silico and biological mock communities containing a limited set of strains, indicating that the pipeline shows the potential to identify strains at species and sometimes subspecies level. In this study, the authors also test their pipeline on a real (biological) mock community and on field samples. In that regard the study has its merits and it could be valuable for other researchers using nanopore sequencing.
However, the discussion of the results is very limited, in particular on the limitations of the platform.
- The difference between amplicon sequencing and metagenome sequencing could be mentioned more clearly in the discussion
- The DNA concentration cannot be accurately measured in the field. What is the impact on sensitivity to detect pathogens in a sample, in particular because in two out of 3 samples only 2 and 3 hits were found (out of 28k or 46k?). It would have been good that the authors could also discuss about the sensitivity of the platform. For instance, can they somehow correlate the number of hits with the number of pathogens present in the sample?
- In addition, it would be good to comment on the limitations of this reduced database in the field (pathogens not in the database are not detected?) and on the impact of the analysis (accuracy and analysis time) if the reference database would be amended with additional pathogens.
- The large discrepancy between the Rapid pipeline run times of the pseudomonas and fusarium mock (table1) should be discussed. Also how is it explained that the run time for fusarium was much shorter, while it contained much more reads? In addition, it would be good to discuss the number of false positives. For instance for the Pseudomonas mock (if I understood correctly), around 5% of the reads are wrongly classified into 83 different taxa
- How could this platform relate to other tools to identify pathogens with regard to specificity and sensitivity? In other words, what could be the value of this platform over other tools.

Additional comments

Although there are several major comments, I would really like to encourage the authors to thoroughly revise the manuscript.

Reviewer 2 ·

Basic reporting

N.A

Experimental design

1. It is unclear whether the final cut-off values for RAPiD are determined automatically or need to be adjusted manually for each dataset. Additionally, how would RAPiD's performance be affected by the ratio of available and unavailable genomes for pathogens within a community? if alignment scores for conserved genes are weighted, would computational accuracy be improved?

2. Does the algorithm include any module that can detect/exclude sequencing contamination from the hosts? How robust the algorithm is in terms of determining sequencing reads from pathogen or host.

Validity of the findings

1. Line 125-127, are each dataset normalized to 100,000 reads or 100.000 reads?
2. For the sequencing data of field-collected samples, 303 sequences were assigned to P. triticina and 3 sequences were assigned to V. inaequalis (Line 312-317), how confident are we about the identity of pathogens?

---

## Round 0.2 · Minor Revisions

Dear Dr. Knobloch and colleagues:

Thanks for revising your manuscript. The reviewers are very satisfied with your revision (as am I). Great! However, there are a few issues to entertain. Please address these ASAP so we may move towards acceptance of your work.

Best,

-joe

·

Basic reporting

Overall I appreciate the efforts of the authors to thoroughly revise manuscript and take into account the major comments.

Experimental design

Lastly in regards to your response to “- Rapid uses a curated database (limited number of sequences in the database and also non-informative (mitochondrial) or contaminated sequences are removed. Considering that most other tools do not make use of a simplified reference database, there may be a bias in the results regarding run time as well as in detection of false-positives. Use of similar databases for all pipelines would give a much better comparison between the pipelines.":
Is there any possibility for the rapid pipeline to be used with e.g. the ncbi nt database, or a really large collection of curated genomes (purely out of interest) despite the much longer runtime of the program that will result from it? (minimap2+megan took around 18:12-24 hours hours, so I guess runtimes of the rapid pipeline would be very similar?) This would be very interesting for microbial community studies where analysis time is less of the essence.

Validity of the findings

Whilst it is regrettable that there was no confirmational test performed on the samples tested in field, the addition of the sequences which were assigned to the supposed taxa identified leave the door open to further analysis if required.
My last question in regards to this matter is if a blast was performed against the ncbi nt database on these sequences, are the identified taxa the only close hits? Were any other species high in their alignment scores, etc. In other words, how certain are you that these sequences really belong to the species.
It would be good to ammend table 6 with highest hits, ID%

Additional comments

L 175: by adding 200 ng gDNA of each organism to a tube.
L204-208: is there anyway to analyze the reads the true positive reads to assess whether they were assigned to the correct species and not the wrong species that is present in the mock as well?
L213-214: What about the dataset that contained 4 pseudomonas (L167-172) strains that were pooled and sequenced together, if one of the reads of this dataset maps to one of the 4 strains, I guess this is counted as a true positive?
L273-275: you mention that a fraction of the retained false positive reads belong to conserved genetic regions, is any action undertaken to remove these false positive reads (e.g. there was mention that mitochondrial genomes were filtered out, how come these still match? Can you possibly just discard rDNA reads, etc?)
L348: “contaminated sequences” instead of “contaminating sequencing”

Reviewer 2 ·

Basic reporting

no comment

Experimental design

no comment

Validity of the findings

no comment

Additional comments

no comment

---

## Round 0.3 · accepted · Accept

Dear Dr. Knobloch and colleagues:

Thanks for revising your manuscript based on the concerns raised by the reviewers. I now believe that your manuscript is suitable for publication. Congratulations! I look forward to seeing this work in print, and I anticipate it being an important resource for groups studying bioinformatics tools for identifying plant microbes. Thanks again for choosing PeerJ to publish such important work.

Best,

-joe